# The Effect of Polymers on Drug Release Kinetics in Nanoemulsion *In Situ* Gel Formulation

**DOI:** 10.3390/polym14030427

**Published:** 2022-01-21

**Authors:** K. Reeta Vijaya Rani, Sruthi Rajan, Mullaicharam Bhupathyraaj, R. Krishna Priya, Nirmala Halligudi, Mohammad Abobakr Al-Ghazali, Sathvik B. Sridhar, Javedh Shareef, Sabin Thomas, Saleem M. Desai, Pandurang D. Pol

**Affiliations:** 1Surya School of Pharmacy, Vikravandi, Villupuram 605652, Tamilnadu, India; reeta_rani07@yahoo.co.in; 2Periyar College of Pharmaceutical Sciences, Tiruchirappalli 620021, Tamilnadu, India; sruthirajan1997@gmail.com; 3College of Pharmacy, National University of Science and Technology, Muscat 130, Oman; nirmalahalligudi@nu.edu.om (N.H.); mohammadalghazali@nu.edu.om (M.A.A.-G.); 4College of Engineering, National University of Science and Technology, Muscat 130, Oman; krishnapriya@nu.edu.om; 5RAK College of Pharmaceutical Sciences, RAK Medical and Health Sciences University, Ras Al Khaimah 11172, United Arab Emirates; sathvik@rakmhsu.ac.ae (S.B.S.); javedh@rakmhsu.ac.ae (J.S.); 6School of Pharmacy, College of Pharmacy & Nursing, University of Nizwa, Nizwa 616, Oman; sabin@unizwa.edu.om; 7Anjuman Arts, Science, Commerce College of PG Studies in English, Vijayapura 586101, Karnataka, India; saleem_m_desai@yahoo.co.in; 8Department of Chemistry, BHS Arts and TGP Science College, Jamkhandi 586103, Karnataka, India; polpandurang@yahoo.in

**Keywords:** Carbopol 934p, glaucoma, Timolol maleate, nanoemulsion, in situ gel

## Abstract

Glaucoma is an ocular condition characterized by elevated intraocular pressure (IOP). Conventional treatments of glaucoma face poor corneal permeability and bioavailability. To address these issues, a nanoemulsion in situ gel of Timolol maleate was developed in this study by adding the polymer Carbopol 934p. Using Carbopol 934p, a novel ophthalmic pH-induced nanoemulsion in situ gel was formulated. The formulation was liquid at pH 4 and quickly gelled when the pH was raised to 7.4 (Lacrimal pH). The pH-triggered in situ gelling mechanism demonstrated continuous drug release over a 24 h cycle. A total of nine trial formulations were prepared (NEI_1_–NEI_9_) and subjected to various physicochemical and in vitro evaluations. According to the in vitro release kinetics, the drug release of Timolol maleate nanoemulsion in situ gel NEI_5_ followed zero-order kinetics, with a release exponent value of 0.902, indicating that the mechanism of release was non-Fickian diffusion regulated. In vivo results showed that Timolol maleate nanoemulsion in situ gel NEI_5_ provided a better-sustained release of the drug, compared with the Timolet OD eye drops. The formulation is stable in storage, with no distinguishable change in appearance, physical properties, quality, and percentage drug release. NEI_5_ also reduces drug administration frequency, which improves patient compliance. Timolol maleate nanoemulsion in situ gel NEI_5_ achieved the goal of controlled drug delivery with extended-release and cost-effectiveness, lowering the dosage and frequency of drug administration, and thus may improve patient compliance. In conclusion, the stable nanoemulsion in situ gel of Timolol maleate NEI_5_ decreases intraocular pressure (IOP) over a prolonged period.

## 1. Introduction

Nanoemulsions are a group of dispersed particles used for pharmaceutical and biomedical aids and vehicles that show great promise for the future of drug therapies, cosmetics, diagnostics, and biotechnologies. Nanoemulsions are defined as oil-in-water (o/w) emulsions, with mean droplet diameters ranging from 50 to 1000 nm [1]. According to the second law of thermodynamics, the o/w nano-sized emulsion is subjected to various instability processes such as aggregation, flocculation, coalescence, Ostwald ripening, and hence eventual phase separation [2]. Unlike thermodynamically stable microemulsions and clear transparent liquid systems, macro (coarse)- and nano-sized emulsions are meta-stable dispersions. However, the stability of the o/w macro (coarse)- and nano-sized emulsions can substantially be improved with the help of suitable emulsifiers or emulators that are capable of forming a mono- or multilayer coating film around the dispersed oil droplets to reduce interfacial tension and to increase droplet–droplet repulsion [3]. 

The proper ratio of oil:water: gum, the appropriate concentration of emulsifying agents, and high-efficiency emulsification equipment that are used to make very low droplet size are the most important factors to develop the o/w nano-sized emulsion with improved stability over the desired period (in comparison with coarse emulsion) can be obtained. The advantages of the nano-sized emulsion system include natural biodegradability, sub-micrometer droplet size range, stabilizability, and substantial drug solubilization either at the innermost oil phase or the o/w interface, minimizing side effects, and improved bioavailability. Due to these advantages, the nano-sized emulsion is now recognized as a promising drug delivery vehicle or carrier for parenteral and topical (ocular and percutaneous) applications [4,5,6].

Initially, in situ gel drug delivery systems are in sol form. There is no gelation process occur. Once administered inside the body through any one of the many routes, such as oral, ocular, rectal, vaginal, injectable, and intraperitoneal routes, the sol form will be converted to gel form due to the gelation process. In ophthalmic products, the formation of viscoelastic gel occurs after the installation of the liquid form of in situ forming hydrogels. These hydrogels are administered through the ocular cul-de-sac route where the hydrogel goes through a phase transition [7].

Natural polymers are mostly used in the preparation of in situ gel dosage form. For example, xyloglucan, a water-soluble anionic polysaccharide of gellan gum and algin are used for ocular drug delivery system.

Many components such as ocular drugs are used to alter the function of the nervous system. Non-steroidal anti-inflammatory drugs are used to prevent growth or to kill microorganisms. The disadvantages of using conventional ocular drug delivery systems such as eye drops are poor bioavailability and poor therapeutic response.

The reason behind these disadvantages is the fast removal of the drug from the eyes due to elevated tear fluids turnover. To overcome these disadvantages, in situ gels are prepared as the ophthalmic dosage form. Sustained drug release is possible from these in situ gels. In situ gels are viscous gels and have longer pre-corneal contact times, compared with conventional eye drops. 

The gellan gum undergoes changeover into the gel state due to the temperature and ionic condition (Ca^++^) in the tear fluid. Due to this property, an aqueous solution of gellan is used in ophthalmic drug delivery.

Glaucoma is a slowly progressive pathology that can result in the loss of peripheral vision, decreased contrast sensitivity, and loss of visual acuity. Due to the asymptomatic nature of the early phases of the disease, most patients experience undiagnosed loss of vision until the advanced stages of the disease have occurred. Thus, the disease is known as the “silent thief of sight”. This indolent optic neuropathy is characterized structurally by a loss of retinal ganglion cells and optic nerve axons. Glaucoma is the second leading cause of the world’s blindness, with nearly 70 million cases worldwide and accounting for 12% of all cases of preventable blindness [8,9,10]. It is estimated that by 2020, close to 4 million Americans will have glaucoma, with 50% undiagnosed and approximately 120,000 individuals developing blindness [11,12].

This work aims to extend drug availability in glaucomatous conditions by adding a different proportion of the polymer Carbopol 934p. Carbopol is a polyacrylic acid polymer, which shows a sol-to-gel transition in an aqueous solution as the pH is raised above its PK_a_ of about 5.5, and it is widely used in ophthalmology to enhance precorneal retention to the eye [13]. Moreover, Carbopol exhibits excellent mucoadhesive properties when compared with other polymers. 

Carbopol 934p is also used in liquid or semisolid pharmaceutical formulations as rheology modifiers. Due to this property of Carbopol 934p, the immediate release kinetics has been modified as zero-order kinetics of the formulation, which provides the more beneficial effect of the gel formulation [14,15]. 

## 2. Materials and Methods

### 2.1. Materials

A gift sample of Timolol maleate (Pure drug, Madras Pharma (P) Ltd., Chennai, India. Castor oil (Lab grade), Tween 80 (Lab grade), benzalkonium chloride, and glycerol (Lab grade) were purchased from Nice Chemicals (P) Ltd., Chennai, India. Potassium dihydro orthophosphate (Lab grade) was purchased from Scientific Chemicals, Chennai, India. Sodium hydroxide (Lab grade) was purchased from Hi Pure Fine Chem Industries, Chennai, India.

### 2.2. Methodology

#### 2.2.1. Formulation of In Situ Gelling System

A conventional emulsion was prepared by dissolving Timolol maleate in castor oil, and glycerol was used as a cosolvent with continuous stirring in a magnetic stirrer. The aqueous solution of Tween 80 and a sufficient amount of water was added and stirred well. The oil phase was added dropwise in continuous phase with stirring at ambient temperature and added the benzalkonium chloride as a preservative. This conventional emulsion was converted into nanoemulsions with the help of a sonication mechanism. The final step was the addition of Carbopol 943p at pH 4. A total number of nine trial batches were prepared for the optimization of process variables [16,17,18]. The process flowchart for nanoemulsion in situ gel is shown in Figure 1.

Different concentrations of emulsifying agents and gelling agents were used in trial batches and studied to have a sustaining effect for 24 h. In all batches, the concentration of drug and oil were kept constant; the data are presented in Table 1.

#### 2.2.2. Characterization

The following parameters were evaluated for all the formulations to confirm the desired release of drug and stability of formulation: visual appearance and clarity, pH, viscosity, gelling capacity, and particle size analysis [19,20,21,22,23]. 

##### Visual Inspection

A visual inspection was carried out behind the dark background to observe the clarity and proper appearance of each formulation. 

##### pH

The pH of each formulation was measured by using a digital pH meter (Elico).

##### Viscosity

A Brookfield viscometer (Brookfield DV-II + Pro viscometer (The Bharat Instruments & Chemicals, Ludhiana, India) with a small sample adapter, having spindle number SC4-18/13R, was used to quantify the viscosity of the prepared nanoemulsions. The gelling property was determined by mixing the 25:7 ratio of the formulation with simulated tear fluid, and the gelation was evaluated by visual examination. The time taken for the formation of gel and the time taken for dissolution was recorded. 

##### Particle Size

Particle size distribution and the average size of particles present in the formula were determined by blue wave analytical mode by DLS method using Zetasize. Figure 2. Atomic force microscopy [24] was used to confirm the size and shape of the particles (Figure 3). 

##### Sterilization and Sterility Testing

Moist heat sterilization is used for killing microorganisms. Autoclaving, as an efficient method to inactivate bacteria, viruses, and other biological material, is recommended for the disposal of regulated medical waste. 

In this study, Timolol maleate nanoemulsion in situ gel was sterilized by moist heat sterilization. This process was carried out at 121 °C for 15 min under pressures of 15 lb/sq. inch. In this process, the moist-heat vapors at high temperatures precipitate or coagulate the cell wall proteins and destroy the microorganisms. The test for sterility is intended for detecting the presence of viable forms of bacteria, fungi, and yeast in sterilized preparations [25,26,27,28]. 

##### Content Uniformity

The vials (*n* = 3) containing the preparation were shaken for 2–3 min, and 100.0 μL of the preparations were transferred aseptically to sterile 25.0 mL volumetric flasks with a micropipette, and the final volume was made with phosphate buffer pH 7.4. The solution was filtered through a 0.45 μm membrane, and the concentration of Timolol maleate was determined at 295 nm, using a double beam UV spectrophotometer [29,30,31,32,33].

##### Compatibility

Drug–excipient compatibility studies were carried out by using Fourier transform infrared (FTIR) spectral analysis. The FTIR absorption spectra of the pure drug and physical admixtures of the drug with various excipients were taken in the range of 400–4000 cm^−1^ using the KBr disc method (Shimadzu IR-Prestige-21) and observed for characteristic peaks of the drug. The FTIR absorption spectra optimized formula is given in Figure 4.

##### In Vitro Release Study

The in vitro dissolution of the prepared in situ gel formulations was performed by diffusion method using an open embedded glass tube. A cellophane membrane pre-soaked in the dissolution media was fixed in the open end of the glass tube, considered as donor compartment that fixed inside 100 mL beaker containing 50 mL of phosphate buffer pH 7.4, which was used as receptor compartment. Then, 1 ml of the preparation was allowed to diffuse via the cellophane membrane to the receptor compartment, which was kept on a magnetic stirrer at 37 °C. Afterward, 5 mL sample was withdrawn in a specified time interval up to 24 h and analyzed by using Shimadzu Double beam UV–Visible spectrophotometer at 295 nm [34]. The cumulative % drug release in all the formulations were given in Table 5 and Figure 5.

##### In Vivo Studies

The intraocular pressure measurement in albino rabbits was studied in Periyar College of Pharmaceutical Sciences, Tiruchirappalli, Tamilnadu, India. (265/1/101/CPCSEA).

##### Intraocular Pressure Studies

The intraocular pressure study was conducted in albino rabbits (Haffkin strain) of either sex weighing between 1.8 kg and 2.5 kg. All experiments were carried out at room temperature [35]. 

Six rabbits were used for this study. Reduction in intraocular pressure (IOP) was measured by Schiotz tonometer. Minimum two readings of IOP were taken before administration of nanoemulsion in situ gel, which was denoted as I_o_. The formulation (0.05 mL) was administered with the help of an insulin syringe in the lower cul-de-sac of one eye.

The control (0.05 mL) was administered in the right eye. Reduction in IOP at time t was denoted as I_t_, and observations were recorded. The graph is plotted as I_n_ versus time where I_n_ = I_t_ − I_o_/I_t_. The same animal was used repeatedly, allowing a minimum of two days between two successive experiments. The results were compared with commercially available Timolet eye drops (containing 0.5% *w/v* of Timolol maleate manufactured by Sun Pharmaceuticals, Chennai, India).

##### Accelerated Stability

Accelerated Stability studies were carried out by exposing NEI_5_ at various temperatures of 40 °C, and 2–8 °C. After a specific period of storage for stability, the in situ gel was evaluated for physical parameters, in vitro drug release, and drug content [36,37,38,39].

## 3. Results

The prepared in situ gel formulations were evaluated for various physicochemical evaluations such as visual appearance, clarity, gelling capacity, viscosity in pH 4 and pH 7.4, particle size, drug content, compatibility, and in vitro diffusion studies. Based on the Physicochemical and in vitro diffusion studies, formulation NEI5 has been selected and subjected to sterilization, in vivo, sterility testing, and accelerated stability studies. There was no microbial growth found for not less than 14 days at 30° to 35 °C in a fluid thioglycollate medium. The intraocular pressure effect of the Timolol maleate nanoemulsion in situ gel was compared with the effect of aqueous Timolet eye drops with 0.5% *w/v*. At 40 °C, there was a slight decrease in the consistency after three months. There was an increase in the viscosity after gelling. Additionally, the gel formed in situ maintained its integrity without dissolving or eroding for a prolonged period. Results are represented in Tables 2–8 and Figures 1 and 2.

### 3.1. Visual Appearance, Clarity, and Gelling Capacity

Evaluation of Visual appearance, Clarity, and Gelling Capacity carried out to find out the physicochemical properties of nine formulations having different compositions.

The status of all the nine formulations in terms of Visual appearance, Clarity, and Gelling Capacity is shown in Table 2.

### 3.2. Evaluation of Viscosity

In order to know the rheological property of the nine formulations in two different pH 4 and, pH 7.4 the viscosity was measured by using A Brookfield viscometer. The values of viscosity are given in Table 3.

### 3.3. Particle Size Analysis

Particle size distribution and the average size of particles was generated by using the formula were determined by blue wave analytical mode by DLS method using Zeta size. As shown in Figure 2, the report showed that the average mean diameter range was 76 nm to 1000 nm.

### 3.4. Atomic Force Microscopy 

In order to confirm the size and shape of the particles, Atomic force microscopy was used. to confirm the size and shape of the particles (Figure 3). The surface morphology analyzed by atomic force microscopy (AFM) result showed a uniform, spherical, and discrete particle without aggregation, which was smooth in the surface and the nanosize range, at 260.4–351.8 nm.

### 3.5. Drug Content

The concentration of Timolol maleate of nine formulations was determined at 295 nm, using a double beam UV spectrophotometer. The percent-age of drug content in all the formulations given in Table 4.

### 3.6. Compatibility Studies

The FTIR absorption spectra optimized formula is given in Figure 4. The FTIR spectra of formulation NEI5 showed that there were no extra peaks other than the normal peak in the spectra of the mixture of the extracts, containing active constituents, and excipients.

### 3.7. In Vitro Diffusion Drug Release Profile

The in vitro dissolution of the prepared in situ gel formulations was performed by diffusion method to find out the release profile of nine formulations. The cumulative % drug release in all the formulations were given in Table 5 and Figure 5. 

### 3.8. In Vivo Studies

Intraocular Pressure is shown is Table 6:

### 3.9. Accelerated Stability Study

Accelerated Stability studies were carried out by exposing NEI5 at various temperatures of 40 °C, and 2–8 °C. After a specific period of storage for stability, the in situ gel was evaluated for physi-cal parameters, in vitro drug release, and drug content.The results are showed in the Table 7 and Table 8. 

## 4. Discussion

In the present investigation, efforts were made to prepare in situ gel of Timolol maleate using surfactant and a gelling agent such as Tween 80 and Carbopol 934p, to enhance drug availability for a prolonged period and hence improve the bioavailability of ocular drugs in glaucomatous conditions. The use of the Carbopol 934p in situ gelling system is sustained by the property of its solutions to transform into stiff gels when the pH is raised. The two main prerequisites of an in situ gelling system are viscosity and gelling capacity. 

To evaluate the rheological behavior, the viscosity of the formulation before and after the pH 4 to 7.4 was evaluated using a Brookfield viscometer. All selected formulations were shear thinning, exhibiting pseudoplastic behavior. All formulations were liquid at room temperature and underwent rapid gelation upon raising the pH 4 to 7.4, with Carbopol 934p formulation showing the optimum variation in viscosity. The comparative rheological properties of NEI_5_ formulations at different pH conditions indicated 129 cps at pH 4 and up to 265 cps at pH 7.4.

The results of visual appearance and clarity, pH, gelling capacity, particle size analysis, and drug content are shown in Table 2, Table 3 and Table 4. The results demonstrate that all prepared formulations had a clear appearance with an acceptable pH and drug content. Moreover, the gelling capacity of NEI_1_, NEI_5_, and NEI_8_ were found to be good, having immediate gelation, and the gel persisted for an extended period.

The particle size result showed that the average mean diameter range was 76 nm to 1000 nm. The surface morphology analyzed by atomic force microscopy (AFM) result showed a uniform, spherical, and discrete particle without aggregation, which was smooth in the surface and the nanosize range, at 260.4–351.8 nm. The FTIR spectra of formulation NEI_5_ showed that there were no extra peaks other than the normal peak in the spectra of the mixture of the extracts, containing active constituents, and excipients, so no evidence was found of interaction with the drug and polymers, and therefore, they are compatible with each other.

Results of in vitro release of NEI_1_–NEI_9_ are illustrated in Table 5 and Figure 5, respectively. The prepared formulations such as NEI_1_, NEI_2_, NEI_4_, and NEI_7_ showed initial burst release. The regression coefficient for Timolol maleate preparation of zero-order plots was found to be 0.923, 0.982, and 0.976 from the NEI_3_, NEI_5_, and NEI_8_. The regression values for the Timolol maleate of first-order plots were found to be 0.905, 0.835, and 0.830 from the NEI_3_, NEI_5_, and NEI8. When the release data were subjected to Higuchi matrix plots, it was observed that formulation for Timolol maleate with regression coefficients of 0.983, 0.990, 0.987 from the NEI_3_, NEI_5_, and NEI_8_ suggested testing diffusion-controlled release. The “n” values obtained the from Korsemeyer–Peppas equation was found to be 0.836, 0.902, 0.839 from the NEI_3_, NEI_5_, and NEI_8_. The diffusion exponent “n” of the Peppas model was more than 0.45, indicating the release of the drug was due to the diffusion (non-fiction) mechanism.

The higher regression coefficient values for each formulation suggested that the formulation NEI_1_–NEI_9_ behaved as matrix types of drug release, with formulation NEI_5_ having the maximum regression value_._ The result showed that formulation NEI_5_ followed zero-order drug release kinetics, which is correlated with the results of the gelling capacity study, proving that NEI_5_ provides immediate gelation for an extended period.

The NEI5 was sterilized by moist heat sterilization. There was no evidence of microbial growth when the formulation NEI5 was incubated for not less than 14 days at 30° to 35 °C in a fluid thioglycollate medium. 

The reduction in the intraocular pressure effect of the Timolol maleate nanoemulsion in situ gel was compared with the effect of aqueous Timolet eye drops with 0.5% *w/v*. The in situ gel formulation greatly reduced the IOP, compared with the marketed conventional formulation.

Accelerated stability testing revealed that the consistency of gel was found to be the same especially at ambient temperature, but at 40 °C, there was a slight decrease inconsistency after three months. Variations were observed in pH values at all storage conditions; the pH of formulations was found to decrease slightly with time. 

The maximum change was observed at 40 °C. It was revealed that fewer changes in drug content and higher drug release were observed when the formulations were stored at refrigerated temperature (2–8 °C). 

## 5. Conclusions

The novel ophthalmic pH-triggered nanoemulsion in situ gel containing Timolol maleate was successfully formulated by using Carbopol 934p. 

The formulation NEI_5_ provided reasonably constant effective levels of drug within the ocular cavity for a period of 24 h, and the in vivo results clearly showed that the Timolol maleate nanoemulsion in situ gel (NEI_5_) provided the best-sustained release of the drug in comparison with the marketed conventional dosage form. Timolol maleate nanoemulsion in situ gel formulation remained stable on storage conditions, with no apparent change in appearance, physical properties, drug content, and percentage drug release. This formulation (NEI_5_) is an alternative to conventional eye drops for improving bioavailability through its longer precorneal residence time and ability to sustain drug release. 

This formulation (NEI_5_) also may reduce the frequency of drug administration, thus improving patient compliance. Timolol maleate nanoemulsion in situ gel (NEI_5_) achieved the objective of controlled drug delivery with prolonged release and cost-effectiveness, which decreases dose and frequency of drug administration and hence can improve patient compliance. In conclusion, the stable nanoemulsion in situ gel of Timolol maleate (NEI_5_) reduces the intraocular pressure over a prolonged period.

## Figures and Tables

**Figure 1 polymers-14-00427-f001:**
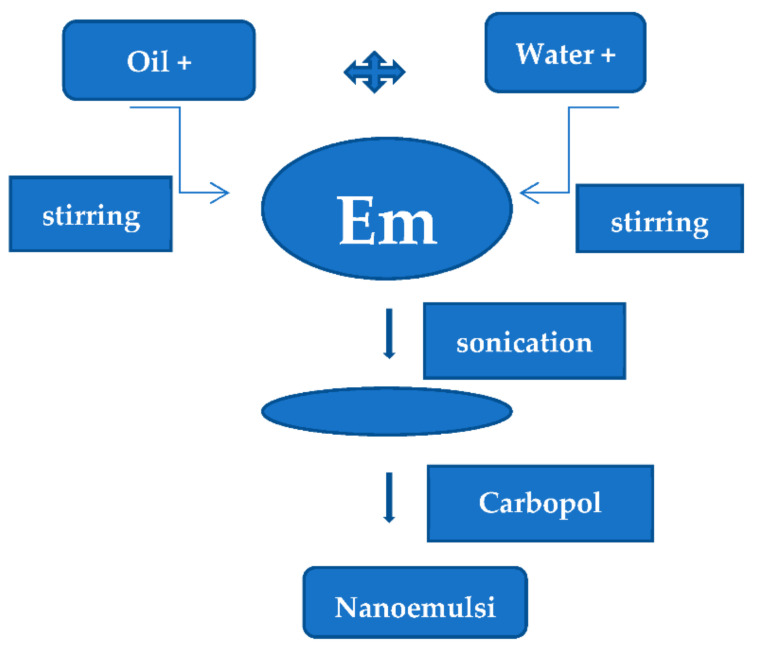
Preparation of nanoemulsion in situ gel by the ultra-sonication method.

**Figure 2 polymers-14-00427-f002:**
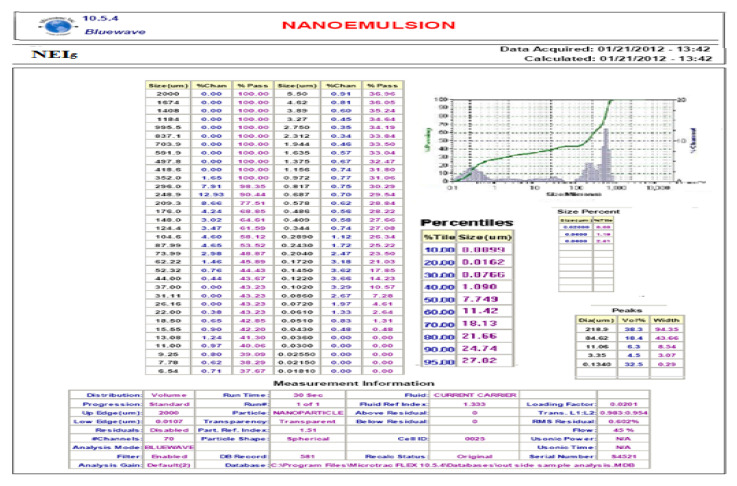
Particle size analysis.

**Figure 3 polymers-14-00427-f003:**
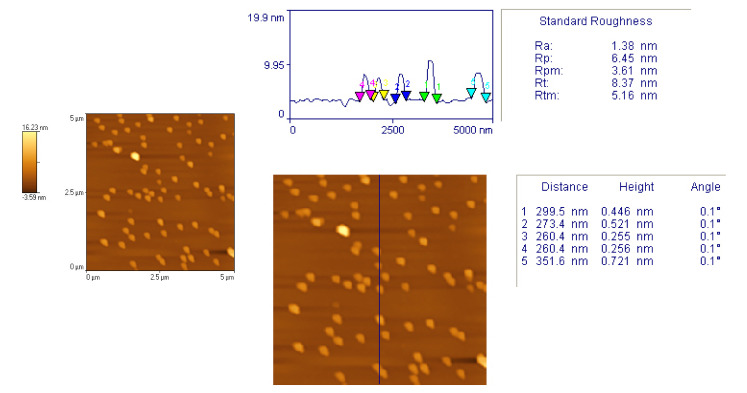
Atomic force microscopy.

**Figure 4 polymers-14-00427-f004:**
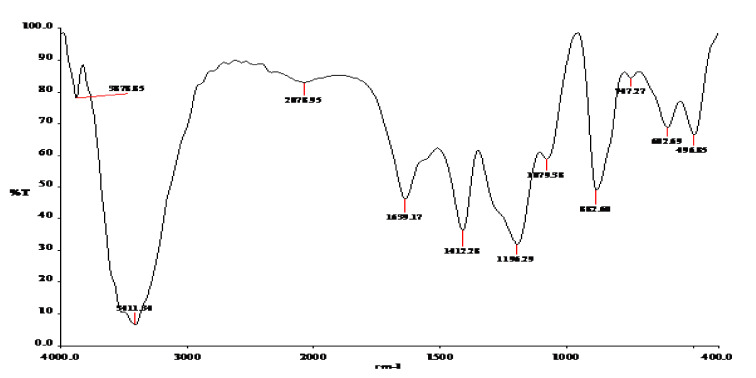
FTIR spectra of NEI5.

**Figure 5 polymers-14-00427-f005:**
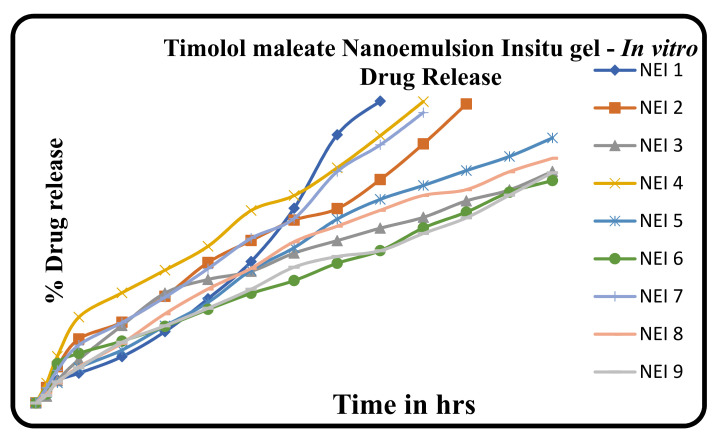
Trail batch. Comparative in vitro diffusion profile of trial formulations.

**Table 1 polymers-14-00427-t001:** Compositions of the trial batch.

Ingredients	NEI_1_	NEI_2_	NEI_3_	NEI_4_	NEI_5_	NEI_6_	NEI_7_	NEI_8_	NEI_9_
Timolol maleate (mg)	100	100	100	100	100	100	100	100	100
Castor oil (mL)	5	5	5	5	5	5	5	5	5
Tween 80 (mL)	3	3	3	3.5	3.5	3.5	4	4	4
Glycerol (mL)	2	2	2	2	2	2	2	2	2
Carbopol 934p	300	600	900	300	600	900	300	600	900
Benzalkonium chloride (%)	0.02	0.02	0.02	0.02	0.02	0.02	0.02	0.02	0.02
Distilled water (mL)	q.s	q.s	q.s	q.s	q.s	q.s	q.s	q.s	q.s

**Table 2 polymers-14-00427-t002:** Visual appearance, Clarity, and Gelling Capacity.

Evaluation	NEI_1_	NEI_2_	NEI_3_	NEI_4_	NEI_5_	NEI_6_	NEI_7_	NEI_8_	NEI_9_
Visual appearance	T	T	T	T	T	T	T	T	T
Clarity	C	C	C	C	C	C	C	C	C
Gelling capacity	+	++	+++	+	++	+++	+	++	+++

T—transparent, C—Clear, + gels slowly and dissolves; ++ gelation immediate and remains for a few hours; +++ gelation immediate and remains for an extended period.

**Table 3 polymers-14-00427-t003:** Evaluation of viscosity in pH 4 and pH 7.4.

Evaluation		Viscosity (cps)								
		NEI_1_	NEI_2_	NEI_3_	NEI_4_	NEI_5_	NEI_6_	NEI_7_	NEI_8_	NEI_9_
pH	4	102	120	140	115	129	143	113	126	140
	7.4	226	260	290	230	265	299	230	260	302

**Table 4 polymers-14-00427-t004:** Drug content.

Formulation	NEI_1_	NEI_2_	NEI_3_	NEI_4_	NEI_5_	NEI_6_	NEI_7_	NEI_8_	NEI_9_
**Drug Content (%)**	101.6	99.56	98.32	100.36	99.79	98.25	100.98	99.43	98.29

**Table 5 polymers-14-00427-t005:** Diffusion profiles of formulations trial batch: NEI_1_–NEI_9_.

Time (min)	Cumulative % Drug Release ± S.D. *								
	NEI_1_	NEI_2_	NEI_3_	NEI_4_	NEI_5_	NEI_6_	NEI_7_	NEI_8_	NEI_9_
0	0	0	0	0	0	0	0	0	0
0.5	5.48 ± 1.1	5.11 ± 0.8	2.26 ± 0.3	6.53 ± 1.1	3.50 ± 0.4	2.97 ± 0.4	4.99 ± 0.3	3.07 ± 0.9	2.51 ± 0.14
1	7.61 ± 1.2	11.95 ± 1.5	7.35 ± 1.1	15.36 ± 1.2	6.65 ± 1.3	12.95 ± 0.2	10.78 ± 0.3	6.66 ± 1.1	6.99 ± 0.38
2	9.83 ± 0.5	21.10 ± 1.4	14.3 ± 2.2	28.31 ± 1.0	11.63 ± 1.3	16.37 ± 0.7	19.14 ± 0.5	11.9 ±0.4	11.59 ± 0.89
4	15.29 ± 0.9	26.63 ±0.7	25.54 ± 2.1	36.33 ± 0.7	17.44 ± 0.9	20.41 ± 1.1	26.52 ± 0.7	19.70 ± 1	20.02 ± 1
6	23.53 ± 0.6	35.17 ± 1.4	36.26 ± 0.09	43.80 ± 1.4	25.44 ± 0.8	25.25 ± 0.9	34.87 ± 0.9	29.42 ± 0.8	25.47 ± 0.96
8	34.37 ± 1.0	46.32 ± 1.0	40.75 ± 1.1	51.68 ± 1.0	33.16 ± 0.4	30.88 ± 0.8	44.30 ± 0.3	37.61 ± 1.1	31.25 ± 1.01
10	46.64 ± 1.1	53.56 ± 1.0	43.43 ± 1.2	63.47 ± 1.3	43.77 ± 0.5	36.16 ± 0.4	54.30 ± 0.9	44.40 ± 1.2	37.53 ± 1.08
12	64.20 ± 1.1	60.30 ± 0.09	49.43 ± 1.4	68.41 ± 1.0	51.23 ± 0.6	40.37 ± 0.4	60.84 ± 0.6	53.19 ± 1.0	44.80 ± 1.08
14	88.45 ± 0.8	64.08 ± 1.7	53.53 ± 0.9	77.54 ± 0.6	60.64 ± 1.2	46.08 ± 0.4	76.31 ± 0.5	58.31 ± 0.9	48.31 ± 0.82
16	99.53 ± 0.9	73.65 ± 1.1	57.65 ± 0.6	88.13 ± 1.8	67.15 ± 0.7	50.24 ± 0.4	85.12 ± 0.6	63.56 ± 0.8	50.22 ± 0.99
18	-	85.40 ± 1.0	61.23 ± 1.6	99.39 ± 1.2	71.95 ± 1.3	57.72 ± 0.9	95.76 ± 0.7	68.46 ± 0.9	55.91 ± 2.46
20	-	98.58 ± 1.1	66.65 ± 1.2	-	77.50 ± 0.3	62.95 ± 1.2	98.72 ± 0.5	70.37 ± 0.8	61.07 ± 0.87
22	-	-	70.28 ± 1.3	-	81.31 ± 0.9	69.50 ± 0.8	-	76.28 ± 1.0	68.54 ± 1.32
24	-	-	76.36 ± 1.32	-	87.40 ± 1.17	73.34 ± 0.7	-	80.71 ± 1.1	75.95 ± 0.48

* S.D. = standard deviation.

**Table 6 polymers-14-00427-t006:** Intraocular pressure (IOP mm Hg).

Time (min)		0	30	60	90	120	150	180	210	240	270	300
Formulation	R	18.7	18.7	18.7	18.7	18.7	18.7	18.7	18.7	18.7	18.7	18.7
	L	23.6	22.3	21.7	17.2	15.4	16.5	21.9	22.3	23.6	23.6	23.6
Marketed	R	18.0	18.0	18.0	18.0	18.0	18.0	18.0	18.0	18.0	18.0	18.0
	L	21.9	21.7	4.3	18.3	19.8	20.8	21.9	21.9	21.9	21.98	21.9

**Table 7 polymers-14-00427-t007:** Stability studies of optimized Timolol maleate nanoemulsion in situ gel.

S. No	Parameters	Initial	After 3 Months	
			40 °C	2–8 °C
2	pH	4	3.9	4
3	Viscosity	129	127	129
4		Drug Content (%)		
1	Timolol maleate	99.79	99.26	99.75

**Table 8 polymers-14-00427-t008:** Comparative in vitro diffusion profiles of Timolol maleate nanoemulsion in situ gel before and after storage at 3 months.

Time (h)	Cumulative % Release		
	Before Storage	After Storage	
		40 °C	2–8 °C
½	3.50	3.05	3.49
1	6.65	6.12	6.50
2	11.63	10.50	11.43
4	17.44	15.52	17.29
6	25.44	24.03	25.26
8	33.16	30.13	32.98
10	43.77	41.52	43.56
12	51.23	49.23	50.98
14	60.64	58.25	60.25
16	67.15	65.50	66.97
18	71.95	69.89	70.79
20	77.50	75.65	77.42
22	81.31	79.23	81.07
24	87.40	83.15	87.35

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
