# Peer review of "The Effect of Polymers on Drug Release Kinetics in Nanoemulsion In Situ Gel Formulation"

_polymers, 2022, doi:10.3390/polym14030427_

Round 1
Reviewer 1 Report
A topical gel forming eye drop of timolol was developed and evaluated in vitro and in vivo. While the authors have presented a significant amount of data, there are some problems with the experimental design and data presentation. Specific comments:
- Introduction is poorly organized. The paragraphs are not logically arranged. Some statements are duplicate (e.g., “Glaucoma is …”).
- Line 43: “Nanoemulsion are thermodynamically stable”. Line 46: “… is subjected to various instability processes”. Line 48: “Unlike microemulsions which are thermodynamically stable and clear”. Please correct the statements.
- The formulations developed were considered nanoemulsions. Which term would be more appropriate, nanoemulsions or microemulsions?
- The authors mentioned many gel forming polymers in the manuscript. However, the authors did not discuss how Carbopol would be better than those polymers.
- The concept of gel forming solution is not new. Timolol gel forming solution is already available in the market. It is not clear why timolol solution, instead of timolol gel forming solution, was used for comparison in the study.
- Figure 1 is not necessary. Please delete.
- Line 200: what was the control in the in vivo studies?
- The authors included all data in the results section but did not describe them.
- It is not necessary to present the same data in both figures and tables. Please delete tables for the same data sets..
- Table7: time should have units in hour. Please check the data for correctness. One of the data was 26.63 +/- 222222222…
- Table 8: there is no unit for time. Which formulation was used for comparison? Please also clarify the formulation in Figure 9.
- Table 10: please clarify which eyes received the developed formulations and which eyes received the marketed eye drops, R or L?
- Figure 11: why no data are presented for the marketed product between 180 and 300 min? Please clarify which formulation was used in this study.
- Table 11: what are the numbers in column 1.
- Line 288: FT-IR analysis was discussed but no data are presented.
- Discussion section is confusing and should be rewritten. The authors only briefly mentioned each study but did not really provide in-depth discussions.
- There are many incomplete sentences and grammatical errors.
Author Response
Review Report Form:
A topical gel forming eye drop of timolol was developed and evaluated in vitro and in vivo. While the authors have presented a significant amount of data, there are some problems with the experimental design and data presentation. Specific comments:
- Introduction is poorly organized. The paragraphs are not logically arranged. Some statements are duplicate (e.g., “Glaucoma is …”).
Answer: Introduction is completely rewritten.
- Line 43: “Nanoemulsion are thermodynamically stable”. Line 46: “… is subjected to various instability processes”. Line 48: “Unlike microemulsions which are thermodynamically stable and clear”. Please correct the statements.
Answer: the statements were corrected.
- The formulations developed were considered nanoemulsions. Which term would be more appropriate, nanoemulsions or micro emulsions?
Answer: nanoemulsions would be more appropriate due to their special purpose. microemulsion is usually prepared to increase the bioavailability, not for sustained release.
- The authors mentioned many gel forming polymers in the manuscript. However, the authors did not discuss how Carbopol would be better than those polymers.
Answer: Page Number 3, Line104, discussed how Carbopol would be better than those polymers.
- The concept of gel forming solution is not new. Timolol gel forming solution is already available in the market. It is not clear why timolol solution, instead of timolol gel forming solution, was used for comparison in the study.
Answer: To assess the sustained release ( zero-order kinetics) property.
- Figure 1 is not necessary. Please delete.
Answer: Deleted
- Line 200: what was the control in the in vivo studies?
Answer: It is understood that the formulation is without a drug.
- The authors included all data in the results section but did not describe them.
Answer: Described in the revised version.
- It is not necessary to present the same data in both figures and tables. Please delete tables for the same data sets..
Answer: Deleted
- Table7: time should have units in hour. Please check the data for correctness. One of the data was 26.63 +/- 222222222…
Answer: Corrected
- Table 8: there is no unit for time. Which formulation was used for comparison? Please also clarify the formulation in Figure 9.
Answer: Corrected
- Table 10: please clarify which eyes received the developed formulations and which eyes received the marketed eye drops, R or L?
Answer: L- for developed formulations and the marketed eye drops of a different set of animals.
R- Control.
- Figure 11: why no data are presented for the marketed product between 180 and 300 min? Please clarify which formulation was used in this study.
Answer: Data included.
- Table 11: what are the numbers in column 1.
Answer: Corrected.
- Line 288: FT-IR analysis was discussed but no data are presented.
Answer: Data included.
- Discussion section is confusing and should be rewritten. The authors only briefly mentioned each study but did not really provide in-depth discussions.
Answer: 16. The discussion section is completely rewritten.
- There are many incomplete sentences and grammatical errors.
Answer: Corrected.
Reviewer 2 Report
- The style of citation in the introduction section is not acceptable. The authors are supposed to add an appropriate reference after each scientific statement. As I can see in the introduction from lines 44-64 (~20 lines) there are no references. Please distribute the references in the text and after the important sentences such as line 45 “According to the second law …. “. There must be a reference here.
- The sentences are too long and full of grammatical errors in the writing. Needs to rewrite the text.
- Again, from lines 74-91, there are no references. Please follow comment number 1 in all the text.
- In the “Materials” section: please bring the specifications of the chemicals including molecular weight, crystallization degree, or any other specifications. How about the purity of the chemicals before using them in the experiments? What is the type of water in the experiments?
- No need to add references in the heading. Remove the references from all the headings and use them in the methodology and discussion. For example, section 2.2.1, 2.2.2, ….
- Line 178, there is a typo-error:” which has been lept in the beaker considered as receptor compartment.”
- Line 184, there is another typo-error:” The results were compared witthe the invitro release study data”.
- Figures 3, and 5 caption is wrong. Please edit the typo error and write the caption with complete details of the graph.
- Figure 4 is not an acceptable figure. Must change and display clear data.
- Figure 6 needs to explain the details of the graph in the caption.
- In the results section, there is no explanation for the results. Please explain completely and compare the results with the recently published references.
- Figures 7, 8, 9, and 12 need error bars.
- I recommend to authors using the following reference in this research:
Sabbagh, F., & Kim, B. S. (2021). Recent advances in polymeric transdermal drug delivery systems. Journal of Controlled Release.
- Replace the too old references with recent and new ones such as reference numbers 8, 14, 15, 16, …. .
Author Response
review Report Form
- The style of citation in the introduction section is not acceptable. The authors are supposed to add an appropriate reference after each scientific statement. As I can see in the introduction from lines 44-64 (~20 lines) there are no references. Please distribute the references in the text and after the important sentences such as line 45 “According to the second law …. “. There must be a reference here.
Answer: All the references included
- The sentences are too long and full of grammatical errors in the writing. Needs to rewrite the text.
Answer: Corrected
- Again, from lines 74-91, there are no references. Please follow comment number 1 in all the text.
Answer: All the references included
- In the “Materials” section: please bring the specifications of the chemicals including molecular weight, crystallization degree, or any other specifications. How about the purity of the chemicals before using them in the experiments? What is the type of water in the experiments?
Answer: All lab-grade chemicals used. Distilled water used.
- No need to add references in the heading. Remove the references from all the headings and use them in the methodology and discussion. For example, section 2.2.1, 2.2.2, ….
Answer: Corrected
- Line 178, there is a typo-error:” which has been lept in the beaker considered as receptor compartment.”
- Answer: Corrected
- Line 184, there is another typo-error:” The results were compared witthe the invitro release study data”.
- Answer: Corrected
- Figures 3, and 5 caption is wrong. Please edit the typo error and write the caption with complete details of the graph.
- Answer: Corrected
- Figure 4 is not an acceptable figure. Must change and display clear data.
- Answer: Corrected
- Figure 6 needs to explain the details of the graph in the caption.
- Answer: Corrected
- In the results section, there is no explanation for the results. Please explain completely and compare the results with the recently published references.
- Answer: Corrected
- Figures 7, 8, 9, and 12 need error bars.
- Answer: Corrected
- I recommend to authors using the following reference in this research:
Sabbagh, F., & Kim, B. S. (2021). Recent advances in polymeric transdermal drug delivery systems. Journal of Controlled Release.vol:341, 132-146
- Replace the too old references with recent and new ones such as reference numbers 8, 14, 15, 16, …. .
Answer: Corrected
Reviewer 3 Report
The introduction of manuscript clearly provided sufficient brief background and included some relevant references. The objectives are clearly stated. The materials and methods have been appropriately described. All tables and very simplified figures are presented in Section 3.Results without any comment. The Fig.4 was not clearly readable. In Section 4.Discussion the numbers of figures and tables have not been indicated, So the presentation of the paper seem to be a bit careless and needs to be improved. More detailed justification in the text of using carbopol could be beneficial.
Author Response
Review Report Form
The introduction of manuscript clearly provided sufficient brief background and included some relevant references. The objectives are clearly stated. The materials and methods have been appropriately described. All tables and very simplified figures are presented in Section
- Results without any comment. Fig.4 was not clearly readable.
Answer: Corrected
In Section 4.Discussion the numbers of figures and tables have not been indicated,
Answer: Corrected
So the presentation of the paper seem to be a bit careless and needs to be improved. More detailed justification in the text of using carbopol could be beneficial.
Answer: Page Number 3, Line104, discussed how Carbopol would be better than those polymers.
Round 2
Reviewer 1 Report
n/a
Reviewer 2 Report
The authors have addressed the comments as well. In the current version of the manuscript, it can be accepted.